# Is publicly-reported firm-level trade data reliable? Evidence from the UK

Holger Breinlich[1], Patrick Nolen[2], Greg C. Wright [3]*

**1** Department of Economics, University of Surrey, CEP and CEPR, Guildford, Surrey, United Kingdom,
**2** Department of Economics, University of Essex, Colchester, United Kingdom, **3** Department of Economics,
University of California, Merced, California, United States of America

* gwright4@ucmerced.edu

**Data Availability Statement:** Data used in this study are third-party data controlled by HMRC. Data are available from the Datalab at HMRC (contact via https://www.gov.uk/government/ organisations/hm-revenue-customs/about/ research#the-hmrc-datalab) for researchers who

## Abstract

In this paper we compare firms' self-reported overseas sales, as reported in a commonly used UK financial reporting dataset, with their actual exports, as reported by Her Majesty's Revenue and Customs (HMRC). Finding that these flows are in several dimensions quite different, we then explore the implications of these differences more formally. Since several studies within the international trade literature report findings based on the self-reported export values in financial datasets, we discuss these findings in light of the departure of financial dataset-based exports from "true" (HMRC) export values.

## 1 Introduction

Firm-level datasets have been increasingly used to explore questions related to international trade. The most common sources for these data are customs records and surveys by national government statistics agencies, while the next most commonly used data come from the self-reports of firms in their end-of-fiscal-year financial reports. These reports are made available to researchers and others by several private data providers, and are available for many countries and regions including the U.S. (CompuStat), the U.K. and Ireland (FAME), Germany (dafne), India (Prowess), the Americas and Asia (Orbis), Russia, Ukraine and Kazakhstan (Ruslana), and Europe (Amadeus), to name just a few. Here we explore the reliability of these financial datasets for international trade research, focusing on the UK and the Bureau van Dijk dataset FAME. In particular, we are motivated by the possibility that these self-reported data may be systematically misreported, which may be very important to the extent that policy decisions are informed by estimates derived from these datasets.

While financial datasets are used as a source of export information for several countries, the UK's FAME database is one of the most widely used. This is in part because UK customs data have only recently become accessible to researchers and the UK's main firm-level production survey (the ABI/ABS) did not contain any information about goods exports until 2011 (and only a binary indicator for export status since then). Researchers have used FAME to explore a range of questions, many of which address fundamental issues within the international trade literature, highlighting the need to understand the extent to which the data accurately reflect the UK economy. Recent work has explored the impact of exporting on R&D [1]; the impact

meet the criteria for access to confidential data. The authors had no special access privileges to the data, and other researchers can obtain the data using the same application process.

**Funding:** The author(s) received no specific funding for this work.

**Competing interests:** The authors have declared that no competing interests exist.

of the financial crisis on exporting [2]; the relationship between the financial health of a firm, exporting, and firm survival [3, 4]; the relationship between exporting and agglomeration economies [5]; the role of exchange rate uncertainty in the export decision [6]; the magnitude of learning-by-exporting [7]; the contribution of exporting to UK productivity growth [8]; the relationship between exporting and firm exit [9]; and firm heterogeneity in barriers to exporting [10]. FAME has also been used extensively for the evaluation of export promotion policies [11, 12].

## 2 Data

We compare patterns of overseas sales across two data sources, restricting our analysis to the manufacturing sector in accordance with most of the literature. The first source is the UK's FAME data, a financial reporting dataset produced by Bureau van Dijk Electronic Publishing, which includes balance sheet information for nearly all UK firms. In addition to reporting a long list of variables related to firm performance and firm finance, FAME also reports "overseas turnover", a variable that primarily captures export sales but also includes the local (overseas) sales associated with the foreign affiliate of a UK firm. This is the variable used as a proxy for export status in the studies listed above and we will refer to it as either "overseas turnover" or simply "exports" throughout this note.

We compare and contrast these FAME-reported values with those reported by another source of export information: the universe of UK transaction-level exports, collected and housed by HMRC. These data are derived from customs declaration forms associated with the physical shipment of goods across borders and should provide a more accurate picture than self-reported exports. In addition, they are not contaminated by the inclusion of local affiliate sales, which are conceptually different from exports.

We merge monthly HMRC transaction-level exports covering the period 2007 to 2010 with FAME, using a common firm identifier. The merged dataset contains two export variables: overseas sales from FAME and data on actual exports from HMRC. Throughout the analysis we also exploit additional firm-level variables such as assets, employment or sales reported by FAME. One issue is that HMRC exports are associated with a trader identification number, which in 26 percent of cases is associated with more than one FAME identifier (HMRC trade flows need to be aggregated to the enterprise group level to be matched to FAME, and these groups often encompass several enterprises). We therefore perform our analysis with a sample that aggregates FAME variables up to the level of each unique trader identification number. We also performed the analysis on the sample of unique FAME-to-HMRC matches (the 74 percent of cases), with very similar results.

## 3 Comparing the FAME and HMRC data—Export status and export values

We begin by asking how well FAME captures some basic facts about export activity. Tables 1 and 2 present an initial comparison of the mean differences in firm activity between exporters and non-exporters in the HMRC data (Table 1) and FAME data (Table 2). We see that firms that report positive exports in FAME are on average larger than the set of exporters in the HMRC data. Figs 1 through 4 provide a more detailed look across the firm size distribution. These Figs illustrate the extent to which FAME self-reported exports deviate from the true distribution of exports by comparing the value of exports and number of exporters reported in both FAME and HMRC, by quartile of firm total assets. Total assets is the only variable available for the universe of firms in FAME and is used as a proxy for size throughout. Fig 1 simply illustrates the fact that there are a greater number of HMRC-reporting (actual) exporters

**Table 1. Summary statistics: HMRC Exporters vs. Non-Exporters.**

| | | No. of Obs. | Mean | Std. Dev. |
|---|---|---|---|---|
| HMRC Exports = 0 | Employees | 19337 | 290 | 2351 |
| | Profits | 56667 | 237881 | 51365600 |
| | Wage Bill | 27616 | 5604663 | 48494461 |
| | Assets | 326310 | 13418688 | 1438800188 |
| | Sales | 51641 | 16372303 | 290379738 |
| HMRC Exports > 0 | Employees | 25957 | 728 | 7424 |
| | Profits | 31366 | 15150697 | 340276404 |
| | Wage Bill | 27604 | 22509585 | 198093817 |
| | Assets | 82852 | 272246942 | 1463492092 |
| | Sales | 28786 | 148204064 | 1621539218 |

relative to FAME-reporting exporters, and this is true for each year in the sample and also true across the firm size distribution. There seems to be a particularly large absence of FAME-reporting exporters among the smallest firms. Fig 2 then narrows the focus to the top percentiles, where the largest disparities are again among the smallest of the large firms. This suggests that the FAME data vastly under-represent the number of exporters in all categories of firm size except for the largest 1 percent. Note that exporters are only required to report intra-EU exports to HMRC if they exceed an annual threshold (£250,000 in 2016). This implies that HMRC might also underestimate the number of actual exporters, suggesting that the disparity between FAME and the true figure may be even greater than reported here.

Across most of the distribution of export volumes there is little difference between FAME-reported values and true HMRC values. However, for the top quartile of firms as measured by assets, FAME-reported export sales vastly overstate both total UK exports as well as the importance of large firms in total exports (Fig 3). Furthermore, Fig 4 shows that the overstatement of exports among large firms in FAME is entirely driven by the concentration of export value among the very largest firms (the top 1 percent). Given the well-documented concentration of large multinational enterprises among the largest firms, the most likely explanation for this pattern is that the inclusion of local affiliate sales in FAME leads to a substantial overestimate of export values at the top of the firm size distribution.

**Table 2. Summary statistics: FAME Exporters vs. Non-Exporters.**

| | | No. of Obs. | Mean | Std. Dev. |
|---|---|---|---|---|
| FAME Exports = 0 | Employees | 26951 | 416 | 6144 |
| | Profits | 67323 | 1918269 | 111409010 |
| | Wage Bill | 36479 | 8825012 | 101148549 |
| | Assets | 388407 | 27036323 | 1867670934 |
| | Sales | 59642 | 35806221 | 673409695 |
| HMRC Exports > 0 | Employees | 18343 | 726 | 5329 |
| | Profits | 20710 | 17361330 | 377189881 |
| | Wage Bill | 18741 | 24235917 | 203480081 |
| | Assets | 20755 | 791797577 | 28685321074 |
| | Sales | 20785 | 143186418 | 1598972829 |

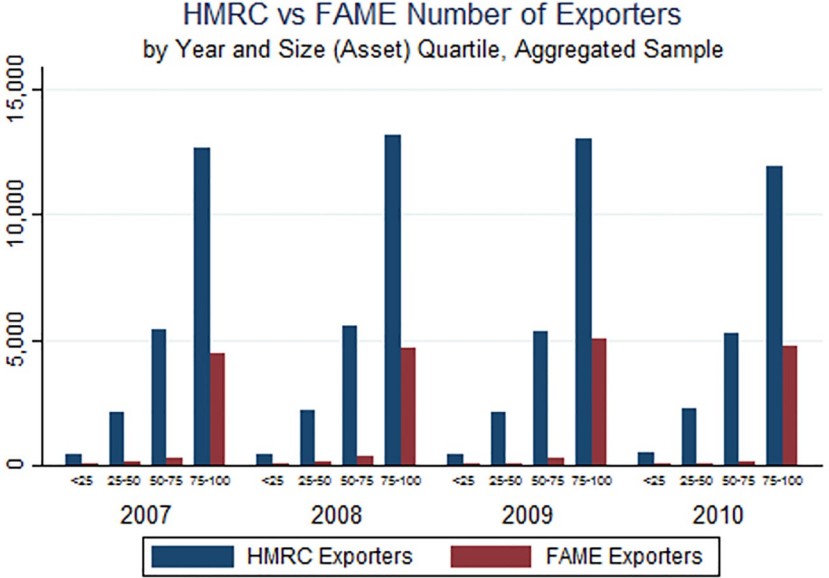

**Fig 1.**

## 4 Implications of mismeasurement—Determinants of export status and exporter premia

While export status and export values are likely to be severely mismeasured in FAME, this does not necessarily invalidate the key results from the studies mentioned earlier. The two principal goals of these and other studies of export behavior are i) to understand the determinants of export status; and ii) to establish whether exporting has a positive and (possibly)

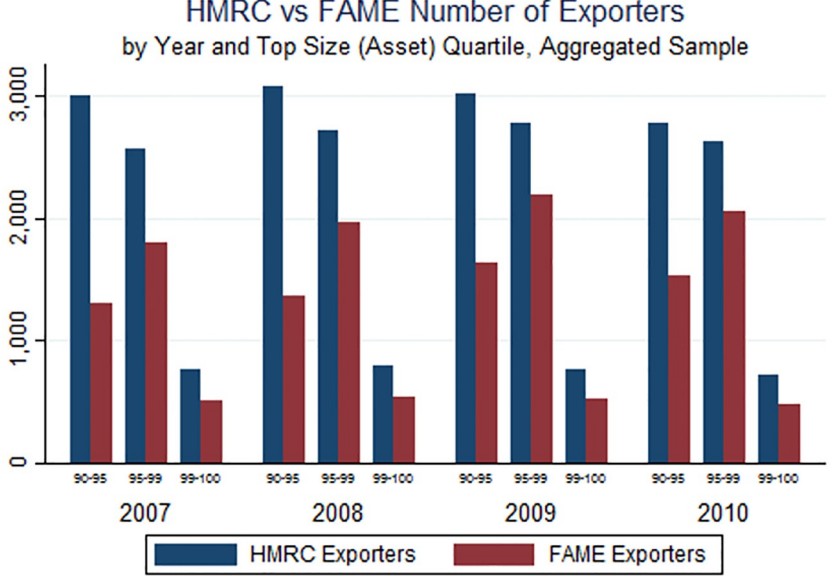

**Fig 2.**

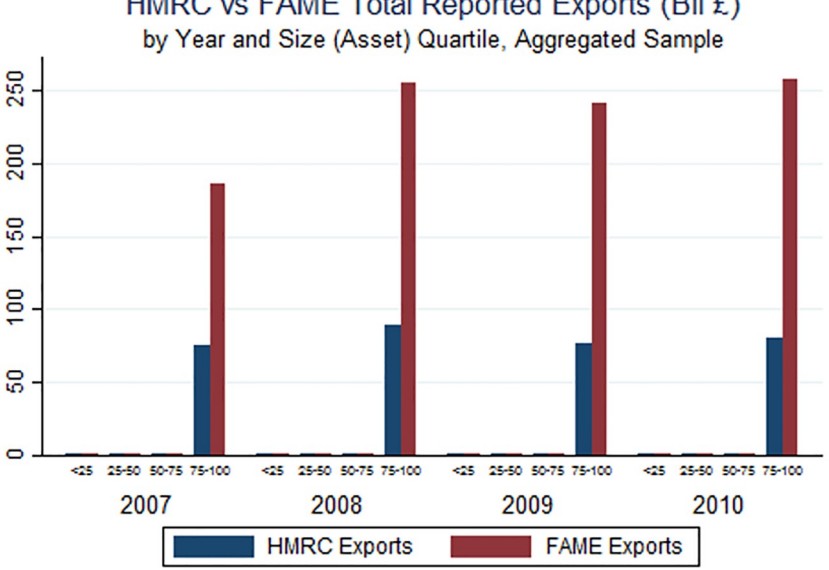

**Fig 3.**

causal association with firm performance indicators ("export premia"). To see whether and how the measurement error introduced by misreporting of exports in FAME changes existing insights, we replicate standard export status and premia regressions for our FAME and HMRC datasets and compare the results. Table 3 reports OLS regression results in which export status (1,0) is regressed on several firm variables. Columns (5)-(8) include year and industry fixed effects while columns (3), (4) and (7), (8) add lagged export status. First, in our preferred

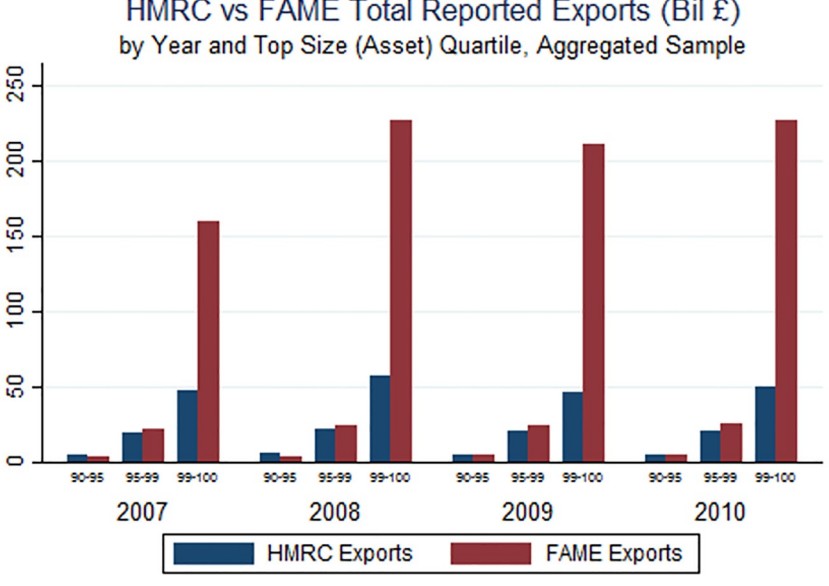

**Fig 4.**

**Table 3. Determinants of export status, FAME vs HMRC.**

| | (1) | (2) | (3) | (4) | (5) | (6) | (7) | (8) |
|---|---|---|---|---|---|---|---|---|
| Dependent variable: *HMRC Exports* > 0 | | | | | | | | |
| Log Total Assets | 0.0714*** | 0.0448*** | 0.0043*** | 0.0031 | 0.0684*** | 0.0487*** | 0.0057*** | 0.0060*** |
| | [0.002] | [0.004] | [0.002] | [0.002] | [0.003] | [0.004] | [0.002] | [0.002] |
| Log Wage Bill | | 0.0397*** | 0.0009 | 0.0092*** | | 0.0361*** | 0.0006 | 0.0101*** |
| | | [0.004] | [0.002] | [0.002] | | [0.003] | [0.002] | [0.002] |
| Log Sales | | | 0.0133*** | | | | 0.0153*** | |
| | | | [0.002] | | | | [0.003] | |
| Lagged Exports | | | 0.8077*** | 0.8137*** | | | 0.7714*** | 0.7765*** |
| | | | [0.008] | [0.008] | | | [0.013] | [0.013] |
| Log Labor Productivity | | | | 0.0137*** | | | | 0.0130*** |
| | | | | [0.002] | | | | [0.003] |
| Industry and Year FE | | | | | | | | |
| Observations | 409,162 | 55,031 | 37,284 | 28,791 | 409,162 | 55,031 | 37,284 | 28,791 |
| R-squared | 0.200 | 0.214 | 0.736 | 0.701 | 0.281 | 0.360 | 0.752 | 0.721 |
| Dependent variable: *FAME Exports* > 0 | | | | | | | | |
| Log Total Assets | 0.0332*** | 0.0191*** | -0.0030* | -0.0044* | 0.0300*** | 0.0214*** | -0.0006 | 0.0007 |
| | [0.002] | [0.004] | [0.002] | [0.002] | [0.002] | [0.003] | [0.002] | [0.003] |
| Log Wage Bill | | 0.0486*** | 0.0223*** | 0.0150*** | | 0.0471*** | 0.0215*** | 0.0155*** |
| | | [0.004] | [0.002] | [0.002] | | [0.003] | [0.002] | [0.002] |
| Log Sales | | | -0.0011 | | | | 0.0009 | |
| | | | [0.002] | | | | [0.002] | |
| Lagged Exports | | | 0.8039*** | 0.8158*** | | | 0.7752*** | 0.7836*** |
| | | | [0.005] | [0.005] | | | [0.006] | [0.006] |
| Log Labor Productivity | | | | 0.0037* | | | | 0.0021 |
| | | | | [0.002] | | | | [0.002] |
| Industry and Year FE | | | | | | | | |
| Observations | 409,162 | 55,031 | 37,284 | 28,791 | 409,162 | 55,031 | 37,284 | 28,791 |
| R-squared | 0.145 | 0.152 | 0.705 | 0.675 | 0.217 | 0.282 | 0.723 | 0.697 |

Robust standard errors in brackets. Dependent variable is Export Status (1,0) in the HMRC or FAME datasets.

*** p<0.01,

** p<0.05,

* p<0.1

specifications, columns (7) and (8), both assets and turnover are positive and highly significant when applied to the HMRC export data, a result that is consistent with the literature. In contrast, these firm size proxies are near zero and not significant when applied to the FAME self-reported exports. And second, the HMRC data show a strong positive relationship between labor productivity and exporting, also consistent with the literature, which is not found in FAME. To summarize, the regressions that adopt the FAME export status variable suggest that firm size and labor productivity play no role in determining whether a firm exports or not. However, the regressions that adopt the HMRC export status variable indicate that larger firms, and more productive firms, are much more likely to export.

In Tables 4 through 6 we estimate export premia by regressing firm capital investment, turn-over, and wages on export status. For each case we estimate OLS specifications with and

**Table 4. Capital export premia, HMRC vs FAME.**

| | (1) | (2) | (3) | (4) | (5) | (6) |
|---|---|---|---|---|---|---|
| Dependent variable: *Log Capital* | | | | | | |
| HMRC Exports > 0 | 2.4451*** | | 0.1681*** | | 0.1013*** | |
| | [0.090] | | [0.016] | | [0.014] | |
| FAME Exports > 0 | | 3.7987*** | | 0.1930*** | | 0.1527*** |
| | | [0.082] | | [0.021] | | [0.020] |
| Log Total Assets | | | 0.9150*** | 0.9207*** | 0.9246*** | 0.9267*** |
| | | | [0.004] | [0.004] | [0.003] | [0.004] |
| Industry FE | | | | | | |
| Year FE | | | | | | |
| Observations | 247,324 | 247,324 | 247,323 | 247,323 | 247,323 | 247,323 |
| R-squared | 0.163 | 0.127 | 0.718 | 0.718 | 0.724 | 0.724 |

Robust standard errors in brackets. Dependent variable is log firm capital investment.

*** p<0.01,

** p<0.05,

* p<0.1

without controls for assets, a proxy for firm size. There is a consistent pattern throughout, namely that the estimates are not very different from one another for both FAME self-reported exports and HMRC exports.

Finally, following the literature we estimate the productivity premia associated with export starters, export stoppers and continuing exporters, for each measure of export status (HMRC versus FAME). Formally, we regress the change in each firm's labor productivity between periods t and t+1 on a set of indicators for whether the firm started exporting, stopped exporting, or continued to export between t and t+1. Table 7 presents the results, where columns (2) and (3) control for firm assets (size) and column (3) also adds industry and year fixed effects. We

**Table 5. Sales export premia, HMRC vs FAME.**

| | (1) | (2) | (3) | (4) | (5) | (6) |
|---|---|---|---|---|---|---|
| Dependent variable: *Log Sales* | | | | | | |
| HMRC Exports > 0 | 3.7092*** | | 0.3973*** | | 0.4191*** | |
| | [0.174] | | [0.036] | | [0.034] | |
| FAME Exports > 0 | | 3.4730*** | | 0.3479*** | | 0.3663*** |
| | | [0.140] | | [0.024] | | [0.023] |
| Log Total Assets | | | 0.8735*** | 0.8836*** | 0.8664*** | 0.8756*** |
| | | | [0.008] | [0.008] | [0.006] | [0.006] |
| Industry FE | | | | | | |
| Year FE | | | | | | |
| Observations | 80,427 | 80,427 | 79,510 | 79,510 | 79,510 | 79,510 |
| R-squared | 0.307 | 0.224 | 0.874 | 0.874 | 0.891 | 0.890 |

Robust standard errors in brackets. Dependent variable is log firm sales.

*** p<0.01,

** p<0.05,

* p<0.1

**Table 6. Wage export premia, HMRC vs FAME.**

| | (1) | (2) | (3) | (4) | (5) | (6) |
|---|---|---|---|---|---|---|
| Dependent variables: *Log Wage* | | | | | | |
| HMRC Exports > 0 | 2.5000*** | | 0.2787*** | | 0.2683*** | |
| | [0.105] | | [0.028] | | [0.024] | |
| FAME Exports > 0 | | 2.2681*** | | 0.3495*** | | 0.3457*** |
| | | [0.103] | | [0.023] | | [0.019] |
| Log Total Assets | | | 0.8525*** | 0.8532*** | 0.8746*** | 0.8744*** |
| | | | [0.009] | [0.009] | [0.010] | [0.009] |
| Industry FE | | | | | | |
| Year FE | | | | | | |
| Observations | 55,220 | 55,220 | 55,031 | 55,031 | 55,031 | 55,031 |
| R-squared | 0.204 | 0.151 | 0.819 | 0.820 | 0.843 | 0.844 |

Robust standard errors in brackets. Dependent variable is the log firm wage bill.

*** p<0.01,

** p<0.05,

* p<0.1

**Table 7. Productivity premium associated with starting, stopping and continuing to export, HMRC vs FAME.**

| | (1) | (2) | (3) |
|---|---|---|---|
| Dependent variable: △ *Labor Productivity* | | | |
| HMRC Exports > 0_start | 0.0119 | 0.0047 | 0.0024 |
| | [0.016] | [0.016] | [0.018] |
| HMRC Exports > 0_stop | -0.0672*** | -0.0731*** | -0.0799*** |
| | [0.017] | [0.017] | [0.019] |
| HMRC Exports > 0_cont | 0.0174** | 0.0078 | 0.0117 |
| | [0.007] | [0.007] | [0.008] |
| Log Total Assets | | 0.0057*** | 0.0085*** |
| | | [0.002] | [0.003] |
| Industry and Year FE | | | |
| Observations | 26,129 | 26,076 | 26,076 |
| R-squared | 0.001 | 0.002 | 0.056 |
| Dependent variable: △ *Labor Productivity* | | | |
| FAME Exports > 0_start | 0.0763*** | 0.0679*** | 0.0626*** |
| | [0.016] | [0.017] | [0.017] |
| FAME Exports > 0_stop | -0.0628*** | -0.0703*** | -0.0758*** |
| | [0.018] | [0.018] | [0.020] |
| FAME Exports > 0_cont | -0.0203*** | -0.0288*** | -0.0336*** |
| | [0.005] | [0.005] | [0.006] |
| Log Total Assets | | 0.0074*** | 0.0111*** |
| | | [0.002] | [0.003] |
| Industry and Year FE | | | |
| Observations | 26,129 | 26,076 | 26,076 |
| R-squared | 0.002 | 0.003 | 0.057 |

Robust standard errors in brackets. Dependent variable is the change in labor productivity between t and t+1. The regressors are indicators (1,0) for whether the firm started exporting, stopped exporting or continued to export between t and t+1.

*** p<0.01,

** p<0.05,

* p<0.1

see that the (negative) premium associated with export stopping is nearly identical in both cases, a result that has also been identified throughout the literature [13]. On the other hand, the true HMRC results suggest no statistically discernible impact of starting or continuing to export, while FAME reports a positive and significant effect of export starting, and a negative and significant effect of continuing to export. The HMRC results are more consistent with the literature (and of course reflect the true behavior of UK firms), which has typically found that firms self-select into exporting, such that the act of beginning to export has little causal impact on productivity levels. With respect to continuing exporters, the evidence from the literature is mixed as to whether there is so-called "learning-by-exporting"—i.e., rising productivity over the export tenure. However, to our knowledge there is no evidence in the literature suggesting that there is a decrease in productivity over the export tenure, as is indicated by the FAME-based result in Column (3).

## 5 Concluding remarks

In this note we have explored the extent to which the export values reported in a widely used U.K. financial dataset, FAME, reflect the true export behavior of those firms. Financial datasets are a commonly used source of export information, and our results should therefore be informative in interpreting existing studies as well as in directing future work that utilizes these data.

Our analysis centers around a comparison of the export values reported in FAME with the true export values collected by HMRC. We conclude with a summary of our findings and some comments on their implications:

- Small (and, possibly, medium-sized) firms often report no exports in FAME when, in fact, they have exported. As a consequence, FAME is unreliable for estimating the total number of exporting firms.

- Export values derived from FAME substantially overstate exports for the largest firms. As a consequence, total exports reported by FAME across industries or economy-wide are not reliable.

- The determinants of export status are not very well captured by FAME. In particular, the relationships between size and exporting, and productivity and exporting, are inconsistent with the HMRC data as well as the existing literature.

- The premia associated with export status are captured fairly well by FAME. One exception is that FAME overestimates the productivity effects associated with starting to export while overstating the losses associated with continuing to export.

## Author Contributions

**Conceptualization:** Holger Breinlich, Patrick Nolen, Greg C. Wright.

**Data curation:** Holger Breinlich, Patrick Nolen.

**Formal analysis:** Holger Breinlich, Patrick Nolen, Greg C. Wright.

**Investigation:** Holger Breinlich, Patrick Nolen, Greg C. Wright.

**Methodology:** Holger Breinlich, Patrick Nolen, Greg C. Wright.

**Project administration:** Greg C. Wright.

**Writing – original draft:** Holger Breinlich, Patrick Nolen, Greg C. Wright.

**Writing – review & editing:** Holger Breinlich, Patrick Nolen, Greg C. Wright.

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
