## [Decision Letter · Decision Letter 0]

7 Jul 2020

PONE-D-20-14506

Is Publicly-Reported Firm-Level Trade Data Reliable? Evidence from the UK

PLOS ONE

Dear Dr. Wright,

Thank you for submitting your manuscript to PLOS ONE. After careful consideration, we feel that it has merit but does not fully meet PLOS ONE’s publication criteria as it currently stands. Therefore, we invite you to submit a revised version of the manuscript that addresses the points raised during the review process.

As you will see in this mail, two experts in international trade provided positive reports for your work with whom I agree. Therefore, I am happy to accept your manuscript for publication in PLOS one. I have only two minor points and hope you can revise them later. First, please confirm the publication year of Greenaway, Kneller and Zhang (2010). It is cited on P.1 but the year is 2008. Second, please put all tables and figures in the manuscript. Thank you for sending them to me by a direct mail, but they are not in the submission system yet.

We look forward to receiving your revised manuscript.

Kind regards,

Dao-Zhi Zeng

Academic Editor

PLOS ONE

Journal Requirements:

4. Please upload copies of Figures (1-4), to which you refer in your text. If the figures are no longer to be included as part of the submission please remove all reference to them within the text.

5. Please include a copies of Tables (1-5) which you refer to in your text.

Reviewers' comments:

Reviewer's Responses to Questions

**Comments to the Author**

1. Is the manuscript technically sound, and do the data support the conclusions?

Reviewer #1: Yes

Reviewer #2: Yes

2. Has the statistical analysis been performed appropriately and rigorously? 

Reviewer #1: Yes

Reviewer #2: Yes

3. Have the authors made all data underlying the findings in their manuscript fully available?

Reviewer #1: No

Reviewer #2: Yes

4. Is the manuscript presented in an intelligible fashion and written in standard English?

Reviewer #1: Yes

Reviewer #2: Yes

5. Review Comments to the Author

Reviewer #1: This note works out a nice idea and presents results that are extremely helpful in putting the results reported in a number of empirical studies for the UK firms and their export activities intp perspective. Imdeed, I now doubt several published results that are published in fine professional journals. I encourage the authors to replicate these studies in detail wuth the "true" export data they have access to and publish their results - in other papers.

Reviewer #2: Using a comprehensive dataset on two types of firm-level trade data, this paper presents an interesting finding on the role of firm-level trade data in empirical trade research, which would help us to interpret the previous findings.

6. PLOS authors have the option to publish the peer review history of their article (what does this mean?). If published, this will include your full peer review and any attached files.

Reviewer #1: **Yes: **Prof. Dr. Joachim Wagner, Leuphana University Lueneburg, Germany

Reviewer #2: No

---

## [Editor Report · Decision Letter 1]

17 Jul 2020

Is Publicly-Reported Firm-Level Trade Data Reliable? Evidence from the UK

PONE-D-20-14506R1

Dear Dr. Wright,

We’re pleased to inform you that your manuscript has been judged scientifically suitable for publication and will be formally accepted for publication once it meets all outstanding technical requirements.

Kind regards,

Dao-Zhi Zeng

Academic Editor

PLOS ONE

Additional Editor Comments (optional):

I have confirmed that the minor points raised in the previous version are corrected. The results are important and interesting so I am happy to accept this version.
---

## [Editor Report · Acceptance letter]

6 Aug 2020

PONE-D-20-14506R1 

Is Publicly-Reported Firm-Level Trade Data Reliable? Evidence from the UK 

Dear Dr. Wright:

I'm pleased to inform you that your manuscript has been deemed suitable for publication in PLOS ONE. Congratulations! Your manuscript is now with our production department. 

Kind regards, 

on behalf of

Dr. Dao-Zhi Zeng 

Academic Editor

PLOS ONE